# Propofol suppresses hormones levels more obviously than sevoflurane in pediatric patients with craniopharyngioma: A prospective randomized controlled clinical trial

Jun Xiong[1], Mengrui Wang[2], Jie Gao[3], Yafen Zhou[4], Yanan Pang[4], Yongxing Sun[4]*

1 Department of Anesthesiology, Shenzhen University General Hospital, Shenzhen University, Shenzhen, Guangdong, 518055, China, 2 Department of Anesthesiology, Peking University Third Hospital, Beijing, 100191, China, 3 Department of Anesthesiology, The First Affiliated Hospital of Kunming Medical University, Kunming, Yunnan Province, 650032, China, 4 Department of Anesthesiology, Sanbo Brain Hospital, Capital Medical University, Beijing, 100093, China

* b2008194@126.com

**Data Availability Statement:** All relevant data are within the paper and its Supporting Information files.

## Abstract

### Objective

General anesthesia can disturb the hormone levels in surgical patients. Hormone deficiency is one of the major symptoms of craniopharyngioma (CP) in pediatric patients. The aim of this prospective randomized controlled clinical study is to evaluate whether propofol and sevoflurane influence the perioperative hormone levels in these patients and to determine which anesthesia technique causes less impact on hormone levels.

### Materials

Sixty-four ASA I and II pediatric patients with CP undergoing elective neurosurgery were randomly divided into the sevoflurane group (S group, n = 32) and the propofol group (P group, n = 32). Anesthesia was maintained with sevoflurane and propofol until the end of the operation. Demographic information, operation information and hemodynamic variables were recorded. The levels of hormones were evaluated preoperatively as the baseline (T0), 1h after the beginning of the operation (T1), immediately at the end of the operation (T2) and 72 h postoperatively (T3).

### Results

There were no significant differences in the two groups in terms of patients' demographics and intraoperative information, such as operation duration, blood loss and transfusion volumes, and fluid infusion volume (*P*>0.05). In both groups, compared to those at T0, the levels of TSH, FT3, TT3 and ACTH at T1, T2 and T3 were significantly lower. The levels of FSH, PRL and GH at T3 were also significantly lower (*P*<0.05). The FT3 and TT3 levels of both groups at T2 and T3 were significantly lower than those at T1, but the ACTH level was

**Funding:** Wu Jieping Medical Foundation of Special Funding Support, No.320.6750.18502. The funders had no role in study design, data collection and analysis, decision to publish, or preparation of the manuscript.

**Competing interests:** The authors have declared that no competing interests exist.

**Abbreviations:** ACTH, Adrenocorticotropic hormone; CP, Craniopharyngioma; FSH, Follicle-stimulating hormone; FT4, Free thyroxine; FT3, Free triiodothyromine; GA, General anesthesia; GH, Growth hormone; LH, Luteinizing hormone; TSH, Thyroid-stimulating hormone; TT4, Total thyroxine; TT3, Total triiodothyronine.

significantly increased ($P<0.05$). Compared to the levels at T2, the TSH, FT3, FT4 and ACTH levels of the two groups at T3 were significantly reduced ($P<0.05$). The baseline hormone levels of both groups were similar ($P>0.05$). At T1, the FT3, TT3, FT4, TT4 and ACTH levels in the P group were significantly lower than those in the S group ($P<0.05$). At T2, the TT3 and ACTH levels of the P group were significantly lower than those of the S group ($P<0.05$) At T3, the TT4 level in the P group was significantly lower than that of the S group ($P<0.05$).

## Conclusion

Propofol and sevoflurane could reduce the levels of hormones intraoperatively and postoperatively in pediatric patients with craniopharyngioma. However, propofol reduced hormone levels more intensively, mainly intraoperatively. Postoperatively, propofol and sevoflurane had similar inhibition effects on the shift in hormone levels. Therefore, in pediatric patients with craniopharyngioma undergoing neurosurgery, sevoflurane might be the preferred anesthetic because it causes less interruption of hormone levels. However, because of their similar postoperative effects, which long-term effects of sevoflurane or propofol could produce optimal clinical situations? Thus more extensive clinical studies are needed.

## Trial registration

**Clinical trial registration.** This trail was registered at Chinese Clinical Trial Registry (http://www.chictr.org.cn, Jun Xiong) on 28/12/2021, registration number was ChiCTR2100054885.

## Introduction

Craniopharyngioma (CP) is a histologically benign neuroepithelial tumor that is located in close proximity to critical neurovascular structures, such as the optic chiasm, hypothalamus and pituitary gland. This rare malformation tumor is most common in children between the ages of 5 and 10 years and is accompanied by endocrine deficiencies, visual impairment and other hormonal symptoms [1,2].

CP may compress adjacent neural structures and impact the pituitary gland's normal function; therefore, primary clinical manifestations of CP are endocrine deficits, including thyroid-stimulating hormone (TSH), adrenocorticotropic hormone (ACTH), growth hormone (GH) and gonadotropins (LH/FSH). The majority of patients with CP present with at least one endocrine deficit [3]. These endocrine deficiencies are also frequently caused by treatment-related lesions to the hypothalamic pituitary axis. Consequently, avoiding irreversible damage to vital neural structures is the key goal in the treatment of CP.

Although neurosurgery and radiation therapy currently provide excellent control for pediatric CP [4], a multidisciplinary approach by a team of endocrinologists, neurosurgeons, neuroanesthesiologists, and neurocritical care specialists can reduce the incidences of postoperative morbidities and late complications associated with the tumor and surgery [5].

It is well known that anesthesia protects patients from experiencing a stress response caused by surgical trauma by suppressing the endocrine stress reaction and sympathoadrenal system [6]. These protective effects are demonstrated by reductions in ACTH and corticol levels in patients undergoing elective surgery [7]. General anesthesia (GA) affects the hypothalamic

pituitary thyroid axis to reduce the serum levels of TSH, free triiodothyromine (FT3) and total triiodothyronine (TT3) [8]. If patients without endocrine disorders, it is indubitable beneficial to reduce these stress hormones with GA. However, is GA advantageous inhibiting hormone levels in pediatric patients with CP?

Anesthesiologists who administer anesthesia to children undergoing resection of CP are concerned with minimizing the interruption of the hypothalamic pituitary axis or the production and secretion of pituitary-derived hormones. Choosing a particular type of anesthetic to produce a preferred pattern of reaction of the endocrine system would be optimally homeostatic and beneficial for these pediatric patients with CP. Unfortunately, there is a shortage of studies comparing the effects of different anesthesia methods on the perioperative levels of hormones, and the issue remains somewhat of a clinical dilemma. Thus, we sought to undertake a study to determine the differences between inhalation anesthetics and total intravenous anesthesia in terms of their impact on hormone levels in pediatric patients with CP. This study would help to assure a high standard of treatment quality.

## Material and methods

### Study

This open prospective randomized blind clinical trial was performed from January to May 2022 at the tertiary care academic teaching hospital after receiving ethics committee approval (SBNK-YJ-2021-030-01). Written informed consent was obtained from the legal guardians of all the pediatric patients. This study was registered before participants were recruited at the Chinese Clinical Trial Registry (http://www.chictr.org.cn, Jun Xiong) on 28/12/2021. The registration number was ChiCTR2100054885. The study was performed according to the guidelines in Ethics Approval and the consent. All methods were conducted in accordance with the Consolidated Standards of Reporting Trials (CONSORT) statement and the Declaration of Helsinki.

### Patients

A total of 64 pediatric patients were approached for inclusion in the study. The inclusion criteria were as follows: Patients with an American Society of Anesthesiologists status I-II undergoing elective CP resection under GA, younger than 18 years, and completion of preoperative measurement of hormones. The exclusion criteria were as follows: patients or their guardians who refused to participate in the study, patients who were suffering from other diseases that caused hormone abnormity or intraoperative accidences possibly interrupting hormone levels, for example, intraoperative massive hemorrhage.

The pediatric patients were divided into two groups at a 1:1 ratio via an SPSS random number generator; the sevoflurane inhalation anesthesia group (S group) and the propofol total intravenous anesthesia group (P group). The allocation was performed by an anesthesiologist who was not involved in the implementation of this study. The group allocation was concealed with sequentially-numbered sealed envelopes and blinded to both the patients and investigators who were responsible for analyzing patient data (Fig 1).

All patients received 2.5 mg dexamethasome and 25 μg levothyroxine orally preoperatively each day. Based on the volume of urine, desmopressin was administered. Moreover, the electrolyte level was regulated to avoid causing major abnormalities. Postoperative hormone supplement treatment with dexamethasone and levothyroxine was routinely administered.

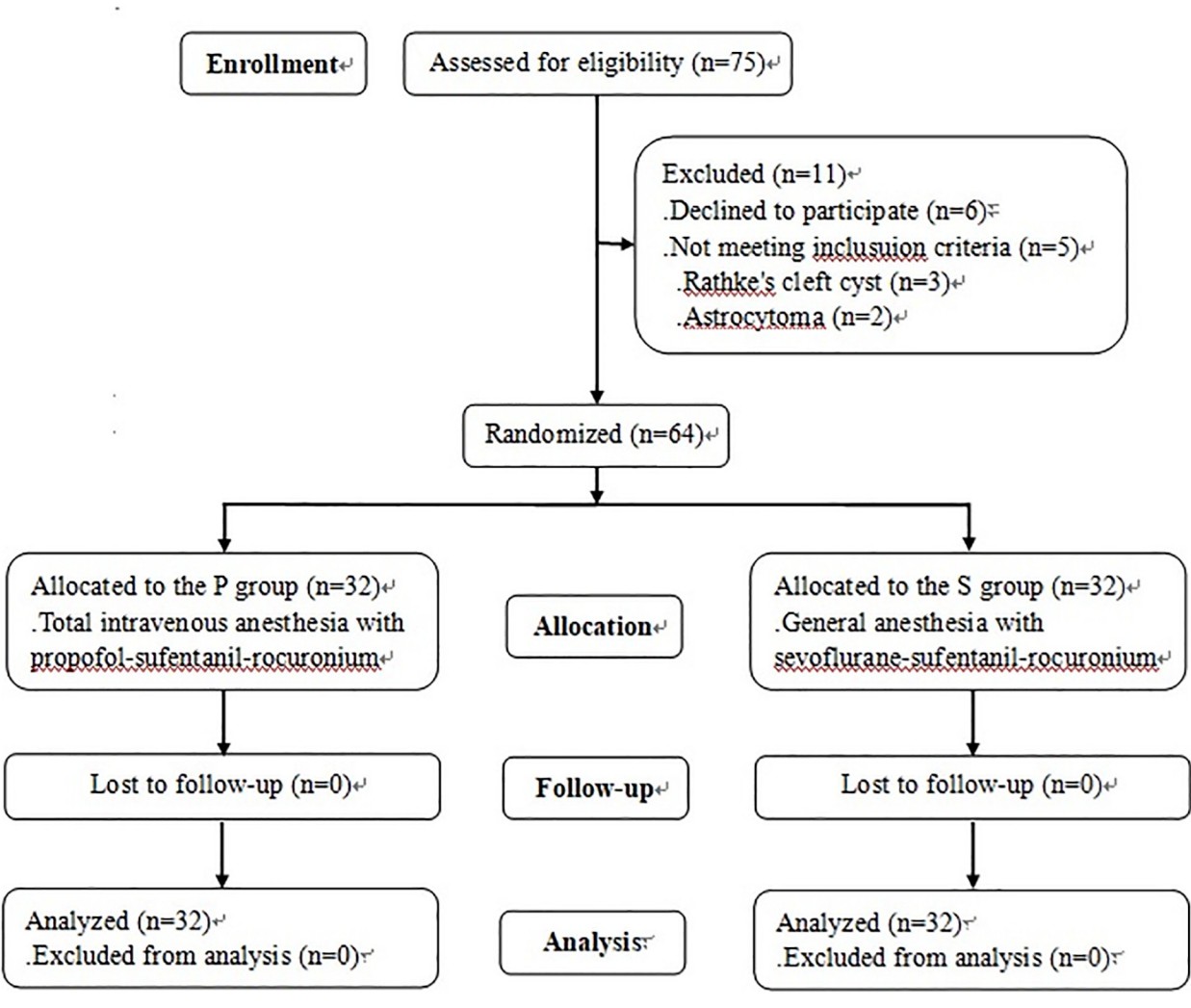

**Fig 1. The CONSORT flow diagram of this study.**

## Study protocol

Pediatric patients' sex, age, weight, and size were measured and recorded by the same health care team. Their body mass index (BMI) was calculated by a monitor (GE CARESCAPE Monitor B650, Helsinki, Finland) based on their demographic variables. All of them obeyed the American Society of Anesthesiologists fasting guidelines. Intravenous access was secured with a cannula in the wards.

After presenting to the operating room, the patients' vital signs, such as heart rate (HR), pulse oxygen saturation ($SpO_2$%), electrocardiogram (ECG) and noninvasive blood pressure (NIBP), were monitored continuously and recorded automatically at 5 min intervals. Entropy index monitoring was performed routinely before GA induction.

GA was induced with 0.03~0.05 mg/kg midazolam, 2~2.5 mg/kg propofol, 0.3~0.4 μg/kg sulfentanil, followed by 0.6~1.0 mg/kg rocuronium. At the same time, facial mask preoxygenation was applied with 100% $O_2$ for 5 min. If hypoventilation or loss of consciousness occurred, ventilation assistance was initiated. After absolute muscle relaxation, endotracheal intubation was completed via a video-laryngoscope. Mechanical ventilation parameters were set with a

tidal volume of 6~10 ml/kg, a 1:2 ratio of inspiration and expiration and a respiratory rate of 10~20 per min to maintain an end tidal pressure of $CO_2$ of 35~45 mmHg. After anesthesia induction, radial arterial cannulation was implemented for continuous arterial pressure monitoring. In addition, an esophageal temperature probe and warm dryer were used to keep the patient's body temperature between 36~37˚C.

Anesthesia was maintained in accordance with different anesthesia protocols. In the S group, patients received sevoflurane, sulfentanil and rocuronium. In the P group, propofol was used instead of sevoflurane. All of these anesthetics were discontinued at the end of the operation. During the operation, the injection speed of anesthetics and concentration of sevoflurane were regulated to keep the entropy index between 40~60, and the shift of mean arterial pressure (MAP) was maintained at less than 20% of the baseline. All patients were operated on the same surgical team with two decades of experience of CP resection.

## Outcomes

The primary outcome measures were the levels of hormones, including glucocorticoid (GC), ACTH, TSH, TT3, TT4, FT3 and FT4. These hormones levels were evaluated preoperatively as baseline levels (T0), 1 h after the beginning of the operation (T1), at the end of the operation instantly (T2) and 72 h after the operation (T3). The secondary outcome measures were operation duration, total fluid infusion volume, urine volume, amount of bleeding, and volumes of allogeneic and autologous transfusion. The MAP and HR upon arrival to the operating room, at the end of the operation, and their peak values were included in the secondary outcome measures. Moreover, the length of hospital stay and the incidence of complications, such as postoperative nausea, cerebral edema, and cerebral or lung infection were also recorded.

## Sample size and statistical analysis

Based on our pilot study, whose protocol was the same as the formal study, FT3 in the S group at 1 h after the beginning of the operation was 0.32±0.28 and that in the P group was 0.15 ±0.14. Thus, the sample size was calculated with a power of 0.9 and alpha error of 0.05. At least 27.2 subjects per group were needed. Assuming a 10% dropout rate and to increase the credibility of this study, 64 patients were recruited (32 subjects for each group).

All statistical analyses were performed with IBM SPSS Statistics V.21.0 (IBM Corp. Beijing, China). Continuous numerical variables were expressed as the mean±standard deviation (SD), and non-normal variables were also reported as the median (range) in addition to the mean ±SD. The normality of these variables was assessed with the Shapiro-Wilk test and histogram. Homogeneity of variance was evaluated by Levene's test, and the means of continuous variables were compared via independent $t$ test or Mann-Whitney $U$ test where appropriate. Qualitative variables were presented as frequencies and percentages, and these variables were evaluated with the Pearson $\chi^2$ test or Fisher's exact test. A $P$ value less than 0.05 was considered statistically significant.

## Results

### General data

A total of 75 pediatric patients with ASA I and II status were enrolled, and 64 participants completed the study. There were 15 (47%) boys and 17 (53%) girls in the S group, and 18 (56%) boys and 14 (44%) girls in the P group. There was no significant differences between the two groups in terms of demographic characteristics or intraoperative information (P>0.05). There were no postoperative cerebral edema or lung complications in the two groups. The

**Table 1. The general characteristics of the two groups.**

|  | S (n = 32) | P (n = 32) |
|---|---|---|
| Age (year) | 8.4±4.6 | 8.3±3.7 |
| Hight (cm) | 124.4±29.5 | 128.6±25.2 |
| Weight (kg) | 34.7±23.6 | 34.4±17.8 |
| BMI (kg/cm$^2$) | 20.1±5.9 | 19.1±4.2, |
| Length of operation (min) | 428.8±122.4 | 427.0±157.7 |
| Allogeneic RBC (ml) | 215.6±188.6 | 209.4±200.6 |
| Autologous RBC (ml) | 111.7±120.0 | 133.8±81.9 |
| Blood loss (ml) | 348.4±218.3 | 423.4±224.3 |
| Fluid infusion volume (ml) | 3806.2±1480.7 | 3641.7±1302.3 |
| Intraoperative urine (ml) | 2106.2±1063.1 | 1829.7±832.1 |
| Postoperative hospital stay (day) | 8.5±4.6 [8 (18)] | 12.5±13.2 [10 (74)] |
| Postoperative nause | 8 (25.0%) | 6 (18.75%) |

S: The S group. P: The P group. BMI: Body mass index. Non-normal variables expressed as the mean±SD and the [median (range)].

incidence of postoperative nausea was also similar (Table 1). Postoperative pathological results demonstrated that the histological subtype of all cases was adamantinomatous CP.

### Hemodynamic variable and perioperative hormone levels of the two groups

Except for the peak MAP of the P group, which was significantly higher than that of the S group ($P = 0.003$), the other hemodynamic parameters were similar (Fig 2).

The perioperative hormone levels of the two groups are shown in Fig 3.

### Comparison of perioperative hormone levels of at different time-points in each group

The hormone levels at the different time points were compared in each group (Table 2). In the S group, compared with those at T0, the levels of TSH, FT3, TT3 and ACTH at T1, T2 and T3 were significantly lower. Levels of FSH, PRL and GH at T3 were significantly lower than those at T0 ($P<0.05$). Compared with the levels at T1, TSH, FT3, TT3, FT4, and TT4 at T2 and T3 were significantly reduced ($P<0.05$). However, the ACTH level at T2 was significantly increased ($P<0.05$). Compared to the levels at T2, the TSH, FT3, FT4 and ACTH levels at T3 were significantly lower ($P<0.05$).

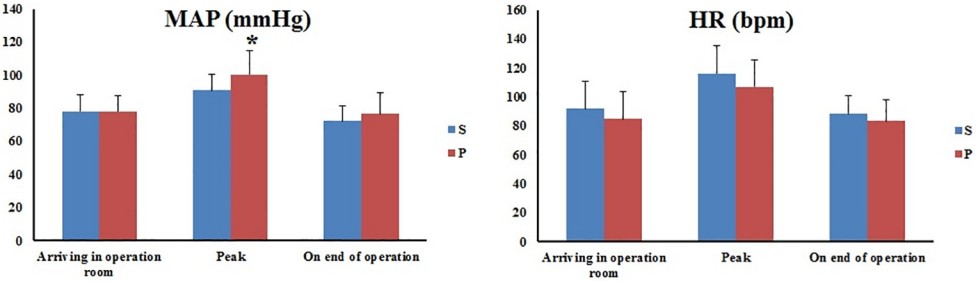

**Fig 2. MAP and HR comparison of two groups.** MAP: Mean arterial pressure. HR: Heart rate. *Compared to S, P<0.05.

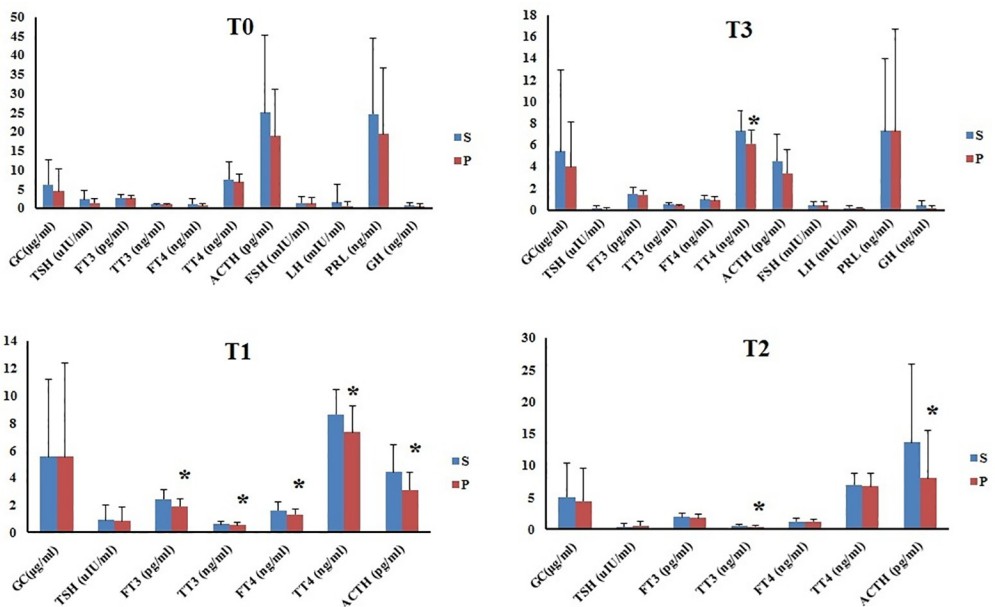

**Fig 3. Perioperative hormones levels of two groups.** S: The S group. P: The P group. *Compared to S, P<0.05. T0: Preoperation. T1: 1h after the beginning of operation. T2: The end of operation. T3: 72h after operation. GC: Glucocorticoid; TSH: Thyroid-stimulating hormone; FT3: Free triiodothyronine; TT3: Total triiodothyronine; FT4: Free thyroxine; TT4: Total thyroxine; ACTH: Adrenocorticotrophic hormone; FSH: Follicle-stimulating hormone; LH: Luteinizing hormone; PRL: Prolactin; GH: Growth Hormone.

In the P group, compared with the levels at T0, the levels of TSH, FT3, TT3 and ACTH at T1, T2 and T3 were significantly reduced (*P*<0.05). The FSH, PRL and GH levels at T3 were also reduced (*P*<0.05). The FT4 levels at T1 and T2 were significantly increased (*P*<0.05). Compared with the levels at T1, the levels of FT3, TT3, and TT4 at T2 and the levels of TSH, FT3, TT3, FT4 and TT4 at T3 were significantly lower (*P*<0.05). However, the ACTH level at T2 was significantly higher than those at T1 (*P*<0.05). Compared to the levels at T2, the levels of TSH, FT3, FT4 and ACTH at T3 were significantly reduced (*P*<0.05).

## Discussion

This study demonstrated that GA impacted on the hormone levels of pediatric patients undergoing CP resection, especially intraoperatively. Compared with sevoflurane, propofol might inhibit the secretion of these hormones more obviously. However, the inhibition effects of the two anesthetics were close to similar level postoperatively. Moreover, intraoperative vital life signs and treatments methods, the incidence of postoperative complications and length of hospital stay were similar between the two anesthesia techniques.

Surgical trauma initiates a neuroendocrine response and is accurately indicated by changes in some neuroendocrine mediators, including ACTH, GC, thyroid hormones, GH and PRL. The stress response can be reduced by GA alone or in combination with other anesthesia techniques [9–11]. This study showed that GA obviously ameliorated the surgical stress response by reducing the levels of these hormones intraoperatively and postoperatively. However, the hormone levels of these pediatric patients undergoing CP resection at postoperative 72 h were still lower than the preoperative levels. These results were significantly different from those of a previous study, in which the levels of ACTH and GC returned close to the baseline levels 72 h postoperatively [12]. We speculated that the lesion and type of operation causing pituitary

**Table 2. The perioperative hormones levels of different time-points in each group respectively.**

| | S (n = 32) | | | | P (n = 32) | | | |
|---|---|---|---|---|---|---|---|---|
| | T0 | T1 | T2 | T3 | T0 | T1 | T2 | T3 |
| GC (μg/ml) | 6.2±6.4[4.4(23.9)] | 5.5±5.7[3.2(21.0)] | 5.0±5.3[3.4(19.8)] | 5.4±7.5[2.3(35.0)] | 4.5±5.8[2.1(25.7)] | 5.5±6.9[2.5(31.3)] | 4.4±5.2[2.4(25.5)] | 4.0±4.1[2.7(15.7)] |
| TSH (uIU/ml) | 2.3±2.4[1.7(8.9)] | 0.9±1.1[0.6(4.1)]* | 0.4±0.5[0.3(1.7)]*& | 0.2±0.2[0.1(1.0)]*&# | 1.4±1.1[1.1(3.9)] | 0.8±1.0[0.4(3.5)]* | 0.5±0.7[0.2(3.5)]* | 0.1±0.1[0.04(0.4)]*&# |
| FT3 (pg/ml) | 2.8±0.8 | 2.4±0.7* | 1.9±0.6*& | 1.5±0.6[1.4(2.9)]*&# | 2.6±0.8 | 1.9±0.5* | 1.7±0.6*& | 1.4±0.4[1.4(2.1)]*&# |
| TT3 (ng/ml) | 1.0±0.3 | 0.6±0.2* | 0.5±0.2[0.5(0.6)]*& | 0.5±0.2[0.4(0.8)]*& | 1.0±0.3 | 0.5±0.2* | 0.4±0.2[0.4(0.6)]*& | 0.4±0.1[0.4(0.6)]*& |
| FT4 (ng/ml) | 1.1±1.4[0.9(8.3)] | 1.6±0.6 | 1.2±0.5& | 1.0±0.3&# | 0.8±0.3[0.8(1.2)] | 1.3±0.4* | 1.2±0.4* | 0.9±0.3&# |
| TT4 (ng/ml) | 7.6±4.5[7.5(28.8)] | 8.6±1.8 | 6.9±1.9[6.9(8.1)]& | 7.3±1.9& | 6.9±2.1[7.0(9.9)] | 7.3±1.9 | 6.7±2.0[6.3(9.7)]& | 6.1±1.3& |
| ACTH (pg/ml) | 25.0±20.3[20.7(74.5)] | 4.4±2.0[4.1(8.5)]* | 13.6±12.2[7.2(37.2)]*& | 4.5±2.5[3.9(10.8)]*# | 18.9±12.2[15.1(49.1)] | 3.1±1.3[3.2(5.5)]* | 8.0±7.4[5.9(26.4)]*& | 3.4±2.2[2.6(8.6)]*# |
| FSH (mIU/ml) | 1.4±1.8[0.7(6.0)] | | | 0.4±0.4 [0.2(1.3)]* | 1.3±1.6[0.3(6.2)] | | | 0.4±0.4[0.2(1.6)]* |
| LH (mIU/ml) | 1.5±4.9[0.2(27.8)] | | | 0.2±0.2[0.2(1.0)] | 0.6±1.2[0.2(5.4)] | | | 0.2±0.0[0.2(0)] |
| PRL (ng/ml) | 24.6±19.8[23.0(99.0)] | | | 7.3±6.7[6.8(26.7)]* | 19.4±17.4[15.6(72.4)] | | | 7.3±9.4[3.9(47.7)]* |
| GH (ng/ml) | 0.7±0.7[0.5(2.9)] | | | 0.4±0.5[0.2(1.9)]* | 0.5±0.6[0.3(1.9)] | | | 0.2±0.2[0.1(0.8)]* |

S: The S group, P: The P group. Non-normal variables expressed as the mean±SD and the [median (range)]

*Compared with T0, P<0.05

&Compared with T1, P<0.05; Compared with T2, P<0.05. T0: Preoperation. T1: 1h after beginning of operation. T2: The end of operation. T3: 72h after operation. GC: Glucocorticoid; TSH: Thyroid-stimulating hormone; FT3: Free triiodothyronine; TT3: Total triiodothyronine; FT4: Free thyroxine; TT4: Total thyroxine; ACTH: Adrenocorticotrophic hormone; FSH: Follicle-stimulating hormone; LH: Luteinizing hormone; PRL: Prolactin; GH: Growth Hormone.

dysfunction might be the main reason for these different results. Additionally, the level of GC was not apparently changed perioperatively in this study. Based on our experience, all pediatric patients with CP should undergo GC supplementation treatment before surgery. This therapy might keep the level of GC stable perioperatively.

CP might compress the pituitary gland and impact its normal function, thereby causing endocrine deficits. Do different anesthetics cause additional injuries in pediatric CP patients who already have endocrine deficits? Although there is no lack of comparative studies of different anesthesia types on the stress response, there are few studies of hormone changes caused by sevoflurane and propofol in pediatric CP patients.

It is known that the choice of anesthetic affects the neuroendocrine stress. In our study, propofol inhibited the levels of ACTH, FT3, TT3, FT4 and TT4 more extensively. At the end of the operation, the levels of TT3 and ACTH were still lower in the propofol group.

GC and ACTH are major biomarkers of the classical stress endocrine response. GA could reduce the stress response induced by surgery. The changes in GC and ACTH levels in our study were consistent with those in a previous retrospective study [13]. Both propofol and sevoflurane inhibited the ACTH level perioperatively, but the effect of propofol was more extensive, and the GC level was much more stable perioperatively. The long-term impact of these anesthetics on GC and ACTH levels was similar in the two groups at 72 h postoperatively. Hyongmin and colleagues demonstrated that there was no significant difference in the effect of both anesthetics on 3-month postoperative neuroendocrine function

in patients who underwent endoscopic transsphenoidal surgery [14]. The results of our study might support the findings of their study. However, in a study of low stress level laparoscopy surgery, propofol not only reduced ACTH level but also cortisol level 30 min after the beginning of surgery [15]. Another similar study also demonstrated that propofol inhibits the ACTH-cortisol axis more extensively than sevoflurane 30 min after surgery onset and at the end of surgery. However, GC concentrations were similar levels in the two groups at 4 h postoperatively [16]. The trend of ACTH shift in our study was similar to that in the above studies, but there were more differences in the reactions of GC to different anesthetics. We speculated that different lesions, different stress levels, and different ages might have influenced the difference. More importantly, these patients received GC supplementation treatment preoperatively, which reduced the shift in the perioperative level of GC. From this evidence, in the presence of surgical stimulation, sevoflurane induced higher ACTH activity than propofol [17]. In other words, propofol might inhibit surgical stress more intensively than inhalation anesthetics [18].

Fewer studies have investigated effects of different anesthetics on thyroid hormone levels. Our results showed that propofol and sevoflurane reduced thyroid hormone levels perioperatively, and the effect of propofol was much stronger. Hypothalamic TSH and other thyroid hormones play several vital roles in human stress [19], which could be obviously inhibited by GA in surgical stimulation. However, a study by Yhim showed that in pituitary adenoma surgery, propofol could better increase the release of TSH and T3 than sevoflurane [13]. And in Marana's study, the effects of propofol on thyroid hormone concentrations were weaker than those of sevoflurane [16]. Based on the present references, we were unable to explain these differences. GA, whether with propofol or sevoflurane, can ameliorate surgical stress reactions, prevent injury, and even cause low T3 syndrome [20]. This was proven with our results because of apparent low levels of thyroid hormones. A study compared the impact of desflurane and propofol on thyroid hormone levels in euthyroid patients. There was no significant difference in the release of TSH, FT3 and FT4 between the two anesthetics [21]. Consequently, which anesthetic induces a more intensive impact on the pituitary thyroid axis should be researched further.

Regarding the antistress effects of GA, we found that propofol and sevoflurane could reduce the levels of PRL and GH 72 h after the operation. However, their influence on GH and PRL levels was similar postoperatively, even at 3 months postoperatively [14].

There is a lack of studies investigating the interruption of different anesthesia techniques on hormone levels in pediatric patients with CP. This is a critical clinical concern because of their hormone deficiencies. Long-term hormone deficiency could lead to retardations in growth and development, an increased risk of cardiovascular disease, and so on [22,23]. If deficiency onset is not only acute but also left untreated, the consequence are life-threating and high mortality, such as the deficiencies in cortisol and TSH. Thus, timely diagnosis and management of this complex condition are important to avoid serious adverse clinical outcomes [24,25]. According to our results, both propofol and sevoflurane demonstrated the ability to reduce hormones levels untill 72 h postoperatively, and the interruption of propofol was more intensive than that of sevoflurane intraoperatively. However, knowledge related to these consequences caused by anesthetic transient inhibition is very limited. Thus, based on our previous experiences and our concern for avoiding unfavorable prognosis, necessary and sufficient hormone replacement was immediately started postoperatively. In fact, the effects of different anesthetics on neuroendocrine function are very complicated, and more extensive clinical studies are needed to confirm the preferred pattern of anesthesia for pediatric CP patients.

## Limitations

In our study, there were still certain limitations to be considered when reviewing the results. First, although this is a prospective randomized controlled study, it is a single-center trial with a small sample size. Consequently, it inevitably lacks external validity. An increase in the sample size might be useful to eliminate or decrease some differences in the results between our study and previous studies. Second, the levels of hormones including PRL, GH, FSH and LH were not evaluated intraoperatively. Third, this study only observed the impact of propofol and sevoflurane on hormone levels 72 h postoperatively, and there has been no long-term follow-up that could reveal more effects of anesthetics on pituitary hormone levels and the prognosis of CP. In our future study, we will increase the sample size and examine these hormones levels intraoperatively. Moreover, a long-term follow-up is needed to examine the effects of hormones supplementation therapy, monitor the hormone levels and determine the incidence of tumor recurrence.

## Conclusion

General anesthesia with sevoflurane and propofol inhibited hormone levels in pediatric patients undergoing CP resection. Propofol might impact these hormone levels more severely during the operation. The inhibiting effects of these anesthetic agents might resemble at 72 h postoperatviely. Sevoflurane might be potent beneficial for these pediatric patients with CP. However, more extensive clinical studies are needed to determine the optimal clinical situation.

## Supporting information

**S1 Checklist. CONSORT 2010 checklist of information to include when reporting a randomised trial\*.**
(DOC)

**S1 File.**
(DOC)

**S2 File.**
(PDF)

**S1 Dataset.**
(XLS)

**S2 Dataset.**
(XLS)

## Author Contributions

**Formal analysis:** Yafen Zhou, Yanan Pang.

**Funding acquisition:** Yongxing Sun.

**Methodology:** Jie Gao.

**Project administration:** Yongxing Sun.

**Writing – original draft:** Jun Xiong, Mengrui Wang.

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
