## [Decision Letter · Decision Letter 0]

10 May 2023

PONE-D-23-08154

Propofol suppresses hormones levels more obviously than sevoflurane in pediatric patients with craniopharyngioma: A prospective randomized controlled clinical trial

PLOS ONE

Dear Dr. Sun,

Thank you for submitting your manuscript to PLOS ONE. After careful consideration, we feel that it has merit but does not fully meet PLOS ONE’s publication criteria as it currently stands. Therefore, we invite you to submit a revised version of the manuscript that addresses the points raised during the review process.

- Registration trial indicated 3*27 patients (3 groups: inhalation, IV, inhalation+IV) whereas 75 were included and 64 analyzed, please reconciliate and provide reason for those discrepancies

- Delete P values for baseline characteristics in Table 1

- Delete t (or T) values in Table 1, 2, and 3

- Please replace at least Table 4 by Figure

- Avoid repetitions in Table and results section

- Please provide a visual abstract

- Check manuscript with a translation service or a native-english speaker

We look forward to receiving your revised manuscript.

Kind regards,

Jean Baptiste Lascarrou

Academic Editor

PLOS ONE

Journal Requirements:

“Wu Jieping Medical Foundation of Special Funding Support, No.320.6750.18502.“

Reviewers' comments:

Reviewer's Responses to Questions

**Comments to the Author**

1. Is the manuscript technically sound, and do the data support the conclusions?

Reviewer #1: Partly

Reviewer #2: Yes

2. Has the statistical analysis been performed appropriately and rigorously? 

Reviewer #1: No

Reviewer #2: Yes

3. Have the authors made all data underlying the findings in their manuscript fully available?

Reviewer #1: Yes

Reviewer #2: Yes

4. Is the manuscript presented in an intelligible fashion and written in standard English?

Reviewer #1: Yes

Reviewer #2: No

5. Review Comments to the Author

Reviewer #1: Sample size calculation:

Need more details about the pilot data. If the pilot is published, need to cite the reference. The means were quite different from this study.

For so many outcomes with a limited sample size, p values should be controlled for family wise error.

Which outcome was compared using the Mann-Whitney test? Seems all were compared with t-tests since mean and SD were presented only.

Mixed model better be used for the repeated measures. It can not only account for within-subject dependence but also test the interaction between treatment and time. A line chart over time is better for readers to understand your data.

Remove t values from the tables.

Reviewer #2: Thank you for the opportunity to review the manuscript by Dr. Xiong and colleagues entitled "Propofol suppresses hormones levels more obviously than sevoflurane in pediatric patients with craniopharyngioma: A prospective randomized controlled clinical trial." A few points of feedback would help strengthen this manuscript.

1) The authors should mention the clinical meaning of any changes in the pituitary hormones measured in the study, and any contextual evidence demonstrating differences in clinical outcome. This will make a stronger case as to why the study is clinically significant.

2) The data in Results, as they are currently presented in a tabular form, are not suitable for publication. Some visual representations of the data - such as bar graphs showing differences in hormone levels between propofol vs. sevoflurane groups - would greatly strengthen the study.

3) The way the manuscript is currently written is difficult to understand and requires should seek independent editorial in order to submit a revision that is written in standard English.

6. PLOS authors have the option to publish the peer review history of their article (what does this mean?). If published, this will include your full peer review and any attached files.

Reviewer #1: No

Reviewer #2: No

---

## [Author Response · Author response to Decision Letter 0]

19 May 2023

Dear editor Jean Baptiste Lascarrou and reviewers:

We sincerely thank you and all reviewers for your valuable feedback. We feel lucky that our manuscript went to these reviewers as the critical comments from them not only help us with the improvement of our manuscript, but also suggest some neat ideas for future studies. Please do forward our heartfelt thanks to these experts. 

Based on all comments we received, careful modifications have been made in the revision manuscript. All changes were marked with red text. In addition, we also asked a native English speaker to check our English writing. We tried our best to revise our manuscript to meet your standard. Below you will find our point-by-point responses to the academic editor and reviewers’ comments or questions:

Response to academic editor’ comments or questions:

1. Registration trial indicated 3*27 patients (3 groups: inhalation, IV, inhalation+IV) whereas 75 were included and 64 analyzed, please reconciliate and provide reason for those discrepancies.

Response: Thank you for your question because it is important. The sample size was calculated based on the pilot study. At least 28 subjects per group were needed. Assuming a 10% dropout rate，that was another 3 subjects. To increase the credibility of this study, we add one subject again, thus there were 32 subjects in each group, and the total cases were 64. When the clinical trial was registered, we forgot the 10% dropout rate. And one author suggested to add the group of inhalation+IV. However, before the onset of the trial, the problem was discussed again. All authors agreed that the trail must obey the primary study protocol and materials of the ethics application. We are very sorry for our negligence to make these discrepancies. However, this did not affect the results and conclusions of our study because it was carried out in accordance with the study protocol and ethics approval. We apologize for our negligence again. In the future, we would compare the effects of sevoflurane, propofol, and sevoflurane+propofol on hormones, which should base on the new registration, study protocol and ethics approval.

2. Delete P values for baseline characteristics in Table 1.

Response: Thanks for your suggestion. We deleted P values in the Table 1. 

3. Delete t (or T) values in Table 1, 2, and 3.

Response: Thanks for your suggestion. We deleted t values in the Table 1, 2 and 3.

4. Please replace at least Table 4 by Figure.

Response: Your advice is constructive really. If that, the figure could give readers directly visual comparison between the results of two groups. However, we tried to make the line graph, the real effect of the line graph was too confused to provide better visual comparison. Because there were too many hormone items in the graph, therefore, each line was too close to another one. The real line graph was as follow. The table 4 demonstrated the detailed values and markers, compared to the line graph, it was better to understand. Actually it is not perfect. However according on your constructive advice, we made bar charts instead of table 2, table 3, and partly table 1. These bar charts was clearly and easily to compare the differences of results. 

5. Avoid repetitions in Table and results section.

Response: Thank you for your valuable suggestion. We revised the part of results to avoid those unnecessary repetitions. 

6. Please provide a visual abstract.

Response: Your suggestion is wonderful because the visual abstract is friendly to reader. This is the first time to provide visual abstract, we try our best to do it. 

Response: We read the submission sample on the web of PLOS ONE’s and try our best to meet the style requirements. If anything did not meet the requirements, we would like very much to modify them. 

8. Thank you for stating the following in the Acknowledgments Section of your manuscript:

“Wu Jieping Medical Foundation of Special Funding Support, No.320.6750.18502.”

Please remove any funding-related text from the manuscript and let us know how you would like to update your Funding Statement. Currently, your Funding Statement reads as follows:“The author(s) received no specific funding for this work.”

Response: Thanks for your reminder. In the manuscript of our submission, we did provide funding information in the part of Funding Statement, but not in the Acknowledgments section. Maybe this negligence took place during online submission. We corrected this mistake during revision submission. Meanwhile, we included our amended statement within the cover letter.

9. We note that you have indicated that data from this study are available upon request. PLOS only allows data to be available upon request if there are legal or ethical restrictions on sharing data publicly. For more information on unacceptable data access restrictions, please see http://journals.plos.org/plosone/s/data-availability#loc-unacceptable-data-access-restrictions.

Response: Thanks for your reminder. Actually, all data of this study are available without any restrictions. Anyone could ask these data from the correspondence author. Meanwhile, we provided the email address of the correspondence author and Data Available Statement in the first submitted manuscript. According on your advice, we would submit these data as Supporting Information files. This problem and the above one (Funding Statement) were negligence caused by unskilled manipulation of online submission system. We are sorry for them.

10. Your ethics statement should only appear in the Methods section of your manuscript. If your ethics statement is written in any section besides the Methods, please delete it from any other section.

Response: According on your advice, we revised to delete it from the part of Title Page.

Response to reviewers’ comment and questions:

Reviewers' comments:

1. Is the manuscript technically sound, and do the data support the conclusions?

Reviewer #1: Partly

Reviewer #2: Yes

Response: Thank you for your earnest comments. We did this clinical trial rigorously as possible as we can, from its design, data collection, data statistical analysis. Although we learned there must be negligence in our study, and it was not perfect, the most inspiring is there is a possibility for us to find the best therapy for our patients. Your valuable suggestions and comments would encourage us to improve in our future studies. 

2. Has the statistical analysis been performed appropriately and rigorously?

Reviewer #1: No

Reviewer #2: Yes

Response: Thanks for your comments. We paid more attention on statistical analysis, including the calculation of sample size and the final data analysis. We did not only consult with statistical expert, but also include an author with statistics background. We tried our best to perform statistical analysis appropriately and rigorously. 

3. Have the authors made all data underlying the findings in their manuscript fully available?

Reviewer #1: Yes

Reviewer #2: Yes

Response: Thank you for your warm comments. All data of this study are available without any restrictions. In the revision submission, we uploaded these data as Supporting Information file. And these could be available from the corresponding author too, the email address as follow: b2008194@126.com.

4. Is the manuscript presented in an intelligible fashion and written in standard English?

Reviewer #1: Yes

Reviewer #2: No

Response: Your comments are really. Although English is not our mother language, we wrote the manuscript clearly and correctly as possible as we can. We also asked a native English speaker to assist our revision. In the revised manuscript, all revisions were marked with red color. We do improve our English writing step by step. 

5. Review Comments to the Author

Reviewer #1: Sample size calculation:

5.1 Need more details about the pilot data. If the pilot is published, need to cite the reference. The means were quite different from this study.

Response: Thank you for your professional comments. Based on the previous references, general anesthesia and surgery affects FT3 significantly, thus it was selected as sample size calculation. The pilot study protocol was as same as the formal one, including general anesthesia method, time-point of blood collection, and so on. However the pilot study was not published because it’s smaller sample size. We also paid close attention on the difference between the means of pilot study and formal study. But we did not forge these data of our study. We are very sorry because of no reasonable explanation for this difference, this study was a real-world study. Maybe the small sample size caused it. In fact, the small sample size was the biggest defect in our study. 

5.2 For so many outcomes with a limited sample size, p values should be controlled for family wise error.

Response: Thank you greatly for your constructive comments. Too small sample size was the biggest defect of our study, which was emphasized in the part of limitation in our manuscript. Thus we provided all related statistical calculation in the manuscript, including t and P values. The limited sample size was our worried shortcoming. However, this study might be the base for future study.

5.3 Which outcome was compared using the Mann-Whitney test? Seems all were compared with t-tests since mean and SD were presented only.

Response: Thanks for your question. In the study, most of hormones levels were compared using Mann-Whitney test because of their abnormal distribution trait. As mentioned above, increasing sample size might make them normal distribution. Because they were continuous numerical variables, therefore they were expressed as mean and SD.

5.4 Mixed model better be used for the repeated measures. It can not only account for within-subject dependence but also test the interaction between treatment and time. A line chart over time is better for readers to understand your data.

Response: Your suggestions were greatly professional. Repeated Measures Anova is an important statistical analysis method. In our study, the mean of two groups, sevoflurane and propofol group, was compared. We focused different anesthetics effects on hormone levels, but not the effect of time on hormone change. Actually, Repeated Measures Anova does be suitable for our study partly, especially on the last part of results (Table 4), that is “The comparison of perioperative hormones levels of different time-points in each group”. Actually, the time-points of blood collection for hormones test weren’t equal in our study, including prioperation, 1h after beginning of operation, the end of operation, and postoperative 72h. The unequal time interval wasn’t as same as standard statistical model of Repeated Measures Anova. In fact, we compared hormones levels of the four time-points mutually in each group. Additionally, we made graphs instead of Table 2, 3, and partly Table 1. As to Table 4, the graph was too confused to be clearly and easily to compare these results because of too many lines of hormones. And these lines were closed to each other. Compared to line graph, Table 4 demonstrated the detailed values and markers. 

5.5 Remove t values from the tables.

Response: According on your advice, we remove t values from the tables. 

Reviewer #2: Thank you for the opportunity to review the manuscript by Dr. Xiong and colleagues entitled "Propofol suppresses hormones levels more obviously than sevoflurane in pediatric patients with craniopharyngioma: A prospective randomized controlled clinical trial." A few points of feedback would help strengthen this manuscript.

1) The authors should mention the clinical meaning of any changes in the pituitary hormones measured in the study, and any contextual evidence demonstrating differences in clinical outcome. This will make a stronger case as to why the study is clinically significant.

Response: Thank you really for your valuable advice. We revised the manuscript to make it clear and easy to understand. Meanwhile, we added new references to show the hazards of hormones deficiency. In other words, it should avoid to disrupt hypothalamic pituitary axis.

2) The data in Results, as they are currently presented in a tabular form, are not suitable for publication. Some visual representations of the data - such as bar graphs showing differences in hormone levels between propofol vs. sevoflurane groups - would greatly strengthen the study.

Response: Your suggestion was constructive. We made three bar charts instead of Table 2, 3, and partly Table 1. These graphs made the manuscript easy to understand. As for the Table 4, the line graph was too confused to provide information clearly since too many hormone lines. Therefore, we gave up the line graph and used the primary table. 

3) The way the manuscript is currently written is difficult to understand and requires should seek independent editorial in order to submit a revision that is written in standard English.

Response: We appreciate your suggestion earnestly. We tried our best to make our English writing clearly and correctly, and asked a native English speaker to assist us. We are now improving our English writing.

On behalf of all the contributing authors, I would like to express our sincere appreciations again. According to these valuable and constructive comments, we have made extensive modifications to our manuscript. If there any other revisions we could make, we would like very much to modify them. We hope the revised manuscript might meet your requirements, though there is a gap between them. Thus, we look forward to your warm comments again. 

Yours sincerely

Correspondence author: Yongxing Sun

Email address: b2008194@126.com

Telephone number: 086-13810197854

---

## [Decision Letter · Decision Letter 1]

2 Jun 2023

PONE-D-23-08154R1Propofol suppresses hormones levels more obviously than sevoflurane in pediatric patients with craniopharyngioma: A prospective randomized controlled clinical trialPLOS ONE

Dear Dr. Sun,

Thank you for submitting your manuscript to PLOS ONE. After careful consideration, we feel that it has merit but does not fully meet PLOS ONE’s publication criteria as it currently stands. Therefore, we invite you to submit a revised version of the manuscript that addresses the points raised during the review process.

 - The manuscript has been improved, even some remarks remained. Please address reviewer's remarks and provide a graphical abstract if possible.

We look forward to receiving your revised manuscript.

Kind regards,

Jean Baptiste Lascarrou

Academic Editor

PLOS ONE

Journal Requirements:

Reviewers' comments:

Reviewer's Responses to Questions

**Comments to the Author**

1. If the authors have adequately addressed your comments raised in a previous round of review and you feel that this manuscript is now acceptable for publication, you may indicate that here to bypass the “Comments to the Author” section, enter your conflict of interest statement in the “Confidential to Editor” section, and submit your "Accept" recommendation.

Reviewer #1: (No Response)

Reviewer #2: (No Response)

2. Is the manuscript technically sound, and do the data support the conclusions?

Reviewer #1: (No Response)

Reviewer #2: Yes

3. Has the statistical analysis been performed appropriately and rigorously? 

Reviewer #1: (No Response)

Reviewer #2: Yes

4. Have the authors made all data underlying the findings in their manuscript fully available?

Reviewer #1: (No Response)

Reviewer #2: Yes

5. Is the manuscript presented in an intelligible fashion and written in standard English?

Reviewer #1: (No Response)

Reviewer #2: No

6. Review Comments to the Author

Reviewer #1: If Mann-Whitney method is used, report median (range) in addition to mean (SD) and put a note on the table.

Reviewer #2: Thank you for the opportunity to review the revised manuscript by Drs. Xiong and colleagues. Please provide additional contextual evidence that more clearly and impact fully describes the clinical significance of decreased pituitary hormones intraoperatively, which is your main outcome finding. Is there any known impact on morbidity and mortality, and clinical outcomes? Is there any clinically meaningful effect of less reduced pituitary hormones in the preoperative period for craniopharyngioma surgery under sevoflurane anesthesia compared to propofol anesthesia? Please comment on this potential impact in the Discussion.

7. PLOS authors have the option to publish the peer review history of their article (what does this mean?). If published, this will include your full peer review and any attached files.

Reviewer #1: No

Reviewer #2: No

---

## [Author Response · Author response to Decision Letter 1]

16 Jun 2023

Dear editor Jean Baptiste Lascarrou and reviewers:

We sincerely thank you and all reviewers for your valuable feedback again. Your comments and suggestions are very useful for improvement of our manuscript revision and future studies. Please do forward our heartfelt thanks to these experts. 

Based on all comments we received, careful modifications have been made in the revision manuscript. All changes were marked with red text. In addition, we invited a native English speaker to check our English writing again. We tried our best to revise our manuscript to meet your standard. Below you will find our point-by-point responses to the academic editor and reviewers’ comments or questions:

Response to academic editor’ comments or questions:

Response: Appreciate you for your academic reminder. We read all references in our manuscript and made sure they were complete and correct. And according with the reviewer’s comment, we also added references in the part of discussion. There was no retracted paper to be cited in our manuscript. 

2. The manuscript has been improved, even some remarks remained. Please address reviewer's remarks and provide a graphical abstract if possible.

Response: Thanks for your comments. During the previous round of revision submission, we already provided a graphical abstract according on your suggestion. 

Response to reviewers' comments and questions:

1. Is the manuscript presented in an intelligible fashion and written in standard English?

Reviewer #1: (No Response)

Reviewer #2: No

Response: Thank you for your comments. We were sorry for this problem because English is not our native language. We tried our best to revise our manuscript and invited a native English speaker to assist our previous round of revision. Although English writing is a difficulty, we have never stopped to improve. We paid more attention on English writing in this revision, and hope our revision to be close to standard English step by step. 

2. Reviewer #1: If Mann-Whitney method is used, report median (range) in addition to mean (SD) and put a note on the table.

Response: Thanks for your suggestion. Non-normal data should be expressed as median (range) or median (quartile). Because these data of hormone levels were continuous numerical variables, so they were expressed as the mean±SD. And this expression style could keep table2 and table3 in the unified form. According with your constructive advice, we checked the primitive statistical data carefully and added the median (range) in addition to the mean (SD). Moreover, these changes were marked on the tables.

3. Reviewer #2: Thank you for the opportunity to review the revised manuscript by Drs. Xiong and colleagues. Please provide additional contextual evidence that more clearly and impact fully describes the clinical significance of decreased pituitary hormones intraoperatively, which is your main outcome finding. Is there any known impact on morbidity and mortality, and clinical outcomes? Is there any clinically meaningful effect of less reduced pituitary hormones in the preoperative period for craniopharyngioma surgery under sevoflurane anesthesia compared to propofol anesthesia? Please comment on this potential impact in the Discussion.

Response: We appreciate your constructive suggestion. Our study did demonstrate propofol and sevoflurane inhibiting pituitary hormones intraoperatively, but what consequences caused by this transient inhibition was litter known. We referred to and increased references related to hypopituitarism in our revised manuscript again. If diagnosis and treatment of hormones deficiency were delayed, clinical prognosis would be bad, even life-threating. Because hormone deficiency could be confirmed and hormone replacement could be immediately started postoperatively in our institution, thus long-term effects of anesthetics inhibiting pituitary hormones on morbidity and mortality of craniopharyngioma would need long-term follow-up. Although a lack of long-term follow-up is one limitation of this study, it is our work in the future study. 

On behalf of all the contributing authors, I would like to express our sincere appreciations again. According to these valuable and constructive comments, we have made extensive modifications to our manuscript. If there any other revisions we could make, we would like very much to modify them. We hope the revised manuscript might meet your requirements, though there is a gap between them. Thus, we look forward to your warm comments again. 

Yours sincerely

Correspondence author: Yongxing Sun

Email address: b2008194@126.com

Telephone number: 086-13810197854

---

## [Decision Letter · Decision Letter 2]

6 Jul 2023

Propofol suppresses hormones levels more obviously than sevoflurane in pediatric patients with craniopharyngioma: A prospective randomized controlled clinical trial

PONE-D-23-08154R2

Dear Dr. Sun,

We’re pleased to inform you that your manuscript has been judged scientifically suitable for publication and will be formally accepted for publication once it meets all outstanding technical requirements.

Kind regards,

Jean Baptiste Lascarrou

Academic Editor

PLOS ONE

Additional Editor Comments (optional):

Reviewers' comments:

Reviewer's Responses to Questions

**Comments to the Author**

1. If the authors have adequately addressed your comments raised in a previous round of review and you feel that this manuscript is now acceptable for publication, you may indicate that here to bypass the “Comments to the Author” section, enter your conflict of interest statement in the “Confidential to Editor” section, and submit your "Accept" recommendation.

Reviewer #1: All comments have been addressed

Reviewer #2: All comments have been addressed

2. Is the manuscript technically sound, and do the data support the conclusions?

Reviewer #1: (No Response)

Reviewer #2: Yes

3. Has the statistical analysis been performed appropriately and rigorously? 

Reviewer #1: (No Response)

Reviewer #2: Yes

4. Have the authors made all data underlying the findings in their manuscript fully available?

Reviewer #1: (No Response)

Reviewer #2: (No Response)

5. Is the manuscript presented in an intelligible fashion and written in standard English?

Reviewer #1: (No Response)

Reviewer #2: Yes

6. Review Comments to the Author

Reviewer #1: (No Response)

Reviewer #2: Thank you for the opportunity to review the second revision of this manuscript submission. The authors have addressed my comments.

7. PLOS authors have the option to publish the peer review history of their article (what does this mean?). If published, this will include your full peer review and any attached files.

Reviewer #1: No

Reviewer #2: No

---

## [Editor Report · Acceptance letter]

20 Jul 2023

PONE-D-23-08154R2 

Propofol suppresses hormones levels more obviously than sevoflurane in pediatric patients with craniopharyngioma: A prospective randomized controlled clinical trial 

Dear Dr. Sun:

I'm pleased to inform you that your manuscript has been deemed suitable for publication in PLOS ONE. Congratulations! Your manuscript is now with our production department. 

Kind regards, 

on behalf of

Dr. Jean Baptiste Lascarrou 

Academic Editor

PLOS ONE